# Design of a Labeling Scheme for 32-QAM Delayed Bit-Interleaved Coded Modulation

**DOI:** 10.3390/s20123528

**Published:** 2020-06-22

**Authors:** Zhe Zhang, Liang Zhou, Junyi Du, Yue Zhao

**Affiliations:** 1National Key Lab on Communication, University of Electronic Science and Technology of China, Chengdu 611731, China; 17781480151@163.com (Z.Z.); zhaoyue@std.uestc.edu.cn (Y.Z.); 2Southwest China Institute of Electronic Technology, Chengdu 610000, China; jydu1989@163.com

**Keywords:** bit-interleaved coded modulation, delayed bit-interleaved coded modulation, gray labeling, depth-first search algorithm

## Abstract

It is very challenging to design the capacity-approaching labeling schemes for large constellations, such as 32-QAM, in delayed bit-interleaved coded modulation (DBICM). In this paper, we investigate the labeling design for 32-QAM DBICM with various numbers of bits delayed by one time slot. In particular, we aim to obtain the labeling schemes with a high DBICM channel capacity by searching the possible labeling schemes. To reduce the search space of the candidate labeling schemes, we propose the criteria that are necessary for good labeling. Based on the proposed criteria, a three-step search algorithm is proposed to obtain the candidate labeling efficiently. Numerical results demonstrate that the DBICM with our proposed labeling scheme can approach the capacity of 32-QAM within 0.015 dB at an information rate greater than 2.5 bits/symbol.

## 1. Introduction

Bit-interleaved coded modulation (BICM) [1] has been widely used in practical systems for its simplicity. It is a bandwidth efficient coded modulation scheme and has been well investigated in point-to-point communications [1,2,3,4,5,6,7]. Compared to trellis-code modulation [8], BICM introduces a random interleaver between the encoder and modulator, which enables researchers to design the encoder and the modulator separately [7]. Therefore, BICM reduces the implementation complexity and attracts enormous attention.

In BICM, the dependence between transmitted bits is ignored by introducing the ideal bit-interleaver. This character is convenient for code design. However, due to the loss of dependence between transmitted bits, the channel capacity of BICM is less than that of coded modulation (CM), which is the constellation capacity. For squared constellations that have Gray labeling schemes, the gap between BICM capacity and constellation capacity is negligible [1]. For non-squared constellations, e.g., eight-quadrature amplitude modulation (QAM) and 32-QAM, Gray labeling is impossible, and the capacity gap is significant. The work [1] proposed the concept of quasi-Gray labeling, and this labeling means to be the best choice for cross constellations. The works [9,10] proposed a design method for quasi-Gray labeling based on the edge-profile. The other newer related works about the labeling focus on the BICM-iterative decoding (ID) [11,12,13,14] or the other constellation [15,16,17,18,19].

To reduce the gap, an improved BICM scheme, namely delayed BICM (DBICM), was proposed in [20]. DBICM maps bits from multiple consecutive codewords to a transmit symbol, which introduces dependence among the codewords. In decoding the consecutive codewords, the decoded result of a codeword is fed back to the demapper as a priori information for the next codeword. Therefore, the DBICM scheme can achieve a considerable gain over BICM. In [20], DBICM was verified to have a higher channel capacity and achieved about 0.7 dB and 0.5 dB gain over the corresponding BICM in eight-amplitude-shift keying (ASK) and half 16-QAM, respectively. It was pointed out in [20] that the labeling scheme is essential to DBICM design, the same as BICM [11,12,15,16,17,18]. However, different from BICM, delay pattern can also affect the performance of DBICM. The DBICM capacity with various delay pattern was investigated in [21]. In [22], the authors proposed the criteria for designing the bit labeling scheme for DBICM with iterative decoding. One of their criteria was based on the harmonic mean of the minimum squared Euclidean distance with a priori information. The other criterion was based on the bit-wise mutual information metrics. Numerical results showed that a 0.5 dB gain was achieved by the optimal labeling scheme designed with their criteria compared to BICM-ID in 16-QAM. A low density parity check (LDPC) code design for DBICM based on the paragraph extrinsic information transfer (EXIT) was proposed in [23]. They achieved a signal-to-noise ratio gain of 0.5 dB to 0.1 dB over the BICM counterparts at a code rate ranging from 0.25 to 0.5. In [24], the authors proposed a pipeline decoding structure of LDPC-DBICM in a two-way relay channel. In the 8-PSK constellation, DBICM with specific delay patterns had about 0.8 dB and 0.35 dB over the BICM scheme, respectively.

Though the DBICM scheme can achieve considerable gain over BICM, it is a great challenge to find the optimal labeling scheme for large constellations, such as 32-QAM. In this work, we attempt to tackle this problem. We consider the cross 32-QAM constellation for its wide application in practical systems and its high energy efficiency [25,26,27,28]. To design the labeling scheme for 32-QAM DBICM in order to maximize the DBICM capacity, a straightforward way is to enumerate all the 32-QAM labeling schemes for each number of delayed bits. However, this is computationally infeasible for the huge number of labeling schemes. Therefore, the search space should be reduced to make the search feasible. Our work focuses on how to reduce the search space and find the good labeling schemes for the 32-QAM constellation. To the best of our knowledge, this is the first work focused on designing labeling for such a large constellation. We summarize the main contributions of this work as follows:

1. We propose three criteria for good labeling schemes that have a high DBICM capacity for 32-QAM based on the Gray labeling rule. In particular, Criterion 1 requires that the partial labels of delayed bits satisfy the Gray labeling rule in the 32-QAM constellation and the partial labels of undelayed bits satisfy the Gray labeling rule in the sub-constellations determined by the values of delayed bits. Criterion 2 requires that both the partial labels of delayed bits and undelayed bits satisfy the Gray labeling rule in the 32-QAM constellation. Criterion 3 requires that 4 bit partial labeling satisfy the Gray labeling in 32-QAM. These proposed criteria significantly reduce the search space for good labeling schemes.

2. We propose an efficient three-step search algorithm to find all candidate labeling schemes based on the proposed criteria. In the first step, we propose a depth-first search algorithm to find the partial labeling based on Criterion 3. Then, we construct the full labeling from that partial labeling. In the end, by eliminating the labeling not satisfying Criterion 1,2, we obtain the candidate labeling. For 32-QAM, the proposed algorithm can find the candidate labeling schemes in a short period of time.

3. We propose to use the sum Hamming distance between all constellation pairs with the minimum Euclidean distance to evaluate the bit error rate (BER) performance of two capacity equivalent labeling schemes. The labeling with a smaller sum Hamming distance is expected to have a better BER performance.

4. Numerical results show that our designed labeling scheme can approach the capacity of 32-QAM within 0.015 dB at an information rate larger than 2.5 bits/symbol. In addition, for the labeling scheme with the same DBICM capacity, the labeling scheme with a smaller sum Hamming distance has a better BER performance than that with a larger Hamming distance.

The remaining parts of this paper are as follows. Section 2 introduces the concept of DBICM. Section 3 analyzes the channel capacity of DBICM and explains why we focus on the delay patterns with various bits delayed by one time slot. Section 4 introduces the proposed criteria for good labeling schemes and the proposed labeling search algorithm. Section 5 demonstrates the designed labeling schemes, the DBICM channel capacity, and the BER results of LDPC coded DBICM schemes. Section 6 draws the conclusion.

## 2. Delayed Bit-Interleaved Coded Modulation

Figure 1 depicts the system model of a DBICM system. At the transmitter side, an information sequence $b_{t}$ of length *k* is encoded and interleaved in time slot *t*. The resultant codeword $c_{t}$ is of length *n*. Then, for a constellation of size $2^{m}$, $c_{t}$ is passed through a one-to-*m* serial-to-parallel converter, leading to *m* length n/m sub-sequences $c_{t}^{i},i=0,\ldots ,m-1$. Here, we assume that *n* is a multiple of *m* for simplicity. All sub-sequences $c_{t}^{i}$ are sent to a bit delay module, in which $c_{t}^{i}$ is delayed by $T_{i}$ time slots. Here, a time slot contains n/m symbol periods. Finally, every *m* bit, in which the $i^{\mathrm{th}}$ bit is drawn from $c_{t-T_{i}}^{i}$, is mapped to a symbol *x* and transmitted through an additive white Gaussian noise (AWGN) channel. In DBICM, the vector $D=\{T_{0},T_{1},\ldots ,T_{m-1}\}$ of length *m* is defined as the delay pattern. It determines the dependence between multiple consecutive codewords.

At the receiver side, noisy signals y are received and fed to the demapper. The demapper calculates the log-likelihood ratios (LLRs) for each bit in $c_{t-T_{i}}^{i}$ based on y and the extrinsic information about the delayed bits of the previous codewords, denoted by $L_{e}\left[c_{t}\right]$, from the decoder. The resultant LLRs are denoted by $L[c_{t-T_{i}}^{i}]$. The LLRs are fed to an inverse bit delay module to reconstruct the LLRs for the codeword $c_{t}$, which are denoted by $L[c_{t}^{i}]$. After an *m*-to-one parallel-to-serial converter and a deinterleaver, the LLRs of the codeword $c_{t}$ are fed to the decoder to obtain the estimated information sequence $\hat{b_{t}}$. The extrinsic information associated with the delayed bits, i.e., $c_{t-T_{i}}^{i},T_{i}\neq 0$, is fed back to the demapper to facilitate the demapping of the undelayed bits, i.e., $c_{t-T_{i}}^{i},T_{i}=0$, of the next codeword.

It can be seen from Figure 1 that two extra modules, a bit delay module and an inverse delay module, are introduced in a DBICM system compared to a BICM system. The bit delay module introduces a coupling effect between consecutive codewords. The receiver explores this coupling effect by demapping the undelayed bits of a codeword with the a priori information associated with the delayed bits from the previous coupled codewords. This leads to a potential SNR gain of DBICM over BICM.

## 3. Channel Capacity Analysis and Delay Pattern Selection

In this section, we analyze the channel capacity of BICM and DBICM. Then, the delay patterns D considered in this paper for DBICM will be specified.

The channel capacity for BICM $C_{BICM}$ is a function of the constellation X, the labeling scheme $L:{\left\{0,1\right\}}^{m}\to X$, and the SNR. In this work, we only consider the cross 32-QAM constellation for a given SNR. Therefore, L determines the BICM channel capacity.

Denote $l\left(x\right):{\left\{0,1\right\}}^{m}$ the label of a constellation point x∈X. Let $l_{i}\left(x\right),i\in \left\{0,1,\ldots ,m-1\right\}$, be the $i^{\mathrm{th}}$ bit of l(x), which is transmitted on the $i^{\mathrm{th}}$ bit channel. Let $I(l_{i};y)$ be the mutual information (MI) between the $i^{\mathrm{th}}$ label bit and the received signal *y*. The BICM capacity for 32-QAM is written as [1]:(1)$$C_{BICM}=\sum_{i=0}^{4}I\left(l_{i};y\right).$$

In BICM, all label bits are considered to be independent. The reason is the introducing of the ideal bit-interleaver between the encoder and the mapper. As a result, the BICM channel capacity is less than the constellation capacity *C* in AWGN channel, which is written as:(2)$$C=I\left(l_{0};y\right)+I\left(l_{1};y|l_{0}\right)+\ldots +I\left(l_{4};y|l_{0},l_{1},l_{2},l_{3}\right).$$

In DBICM, the channel capacity is not only a function of the constellation X, the labeling scheme L, and the SNR. It is also a function of the delay pattern D. Denote the indexes of the delayed bits, i.e., $l_{i}$ with $T_{i}\neq 0$, by A and the indexes of the undelayed bits, i.e., $l_{i}$ with $T_{i}=0$, by $A^{c}=\left\{0,\ldots ,m-1\right\}\setminus A$. As explained in Section 2, the demapping of delayed bits $l_{A}$ only depends on the received signals, and therefore, the bit channel capacities for these bits are the same as those in BICM. However, the demapper utilizes the a priori information associated with the delayed bits in previous codewords to calculate the LLRs of the undelayed bits $l_{A^{c}}$. Therefore, the bit channel capacities of the undelayed bits are different from those in BICM. In the following, we show two example delay patterns in Figure 2. By assuming that the delayed bits from previous codewords are perfectly known by the demapper, the DBICM capacities of these delay patterns are calculated.

In Figure 2a, $l_{0}$ is delayed by one time slot. For this delay pattern, the DBICM channel capacity $C_{DBICM}$ is written as:(3)$$C_{DBICM}=I\left(l_{0};y\right)+\sum_{i=1}^{4}I\left(l_{i};y|l_{0}\right).$$

In Figure 2b, $l_{i}$ is delayed by m−i−1 time slots. In this case, the DBICM channel capacity $C_{DBICM}$ is written as:(4)$$\begin{array}{cc} \; & C_{DBICM}\\ \; & =I\left(l_{0};y\right)+I\left(l_{1};y|l_{0}\right)+\ldots +I\left(l_{4};y|l_{0},l_{1},l_{2},l_{3}\right). \end{array}$$

It can be seen from Equation (Equation 4) that the DBICM capacity for the delay pattern in Figure 2b is equal to the 32-QAM constellation capacity *C* (as shown in Equation (Equation 2)) regardless of the labeling scheme. Therefore, it is optimal in the sense that it maximizes the DBICM capacity. However, this delay pattern has a large decoding latency since the codeword transmitted in time slot *t* cannot be decoded until time slot t+4. In this paper, we consider the delay patterns in which the delayed bits $l_{A}$ are delayed by only one time slot, i.e., $T_{i}\leq 1$ for i∈{0,…,m−1}. This minimizes the decoding latency introduced by DBICM. More importantly, we will show in Section 5 that by carefully designing the labeling scheme for this type of delay patterns, the DBICM capacity can almost achieve the constellation capacity at a moderate to high information rate. For this type of delay patterns, the DBICM capacity is written as:(5)$$\begin{array}{c} C_{DBICM}=\sum_{i\in A}I\left(l_{i};y\right)+\sum_{j\in A^{c}}I\left(l_{j};y|l_{A}\right). \end{array}$$

## 4. Design of the Labeling Scheme

In this section, we consider the design of labeling schemes for the delay patterns of interest in this work. In particular, we aim to maximize the DBICM capacity by designing labeling schemes and their associated delay patterns. To achieve this, a straightforward way is to enumerate all possible combinations of the considered delay patterns and the labeling schemes and calculate the capacity for each combination. The one that maximizes the DBICM capacity is selected as the optimized labeling scheme and the associated delay pattern. However, this is computationally infeasible since there are $P_{32}^{32}$ labeling schemes for 32-QAM. Here, $P_{n}^{m}=n!/\left(n-m\right)!$ represents the permutation of *m* elements from *n* elements. By considering the delay patterns, the number of combinations is even larger.

In the following, we propose the criteria that should be satisfied by a good labeling scheme for a given delay pattern. Based on the proposed criteria, the size of the candidate labeling set will be small enough. Then, we propose an efficient three-step search algorithm, which utilizes the proposed criteria to obtain the candidate labeling. After we eliminate some equivalent labeling from the candidate set, we enumerate all the candidate labeling and choose the one with the highest DBICM capacity and the best BER performance as our designed labeling.

### 4.1. Labeling Search Criteria

To present the search criteria, let us first introduce the concepts of the partial label (PL), partial labeling scheme (PLS), and partial Gray labeling scheme (PGLS).

#### 4.1.1. Partial Gray Labeling Scheme

**Definition** **1.**
***(PL)***
*: For a given non-empty index set $C=\{c_{j}|c_{j}\in \left\{0,1,\ldots ,m-1\right\}\}$, the PL of a constellation point x is defined as l(x) excluding $\left\{l_{j}\left(x\right),j\in \bar{C}\right\}$, which is denoted by $l^{C}\left(x\right)$. Here, $\bar{C}$ is the complementary set of C in {0,1,…,m−1}, i.e., $\bar{C}=\left\{0,1,\ldots ,m-1\right\}\setminus C$.*


**Definition** **2**
***(PLS)***
*: For a given non-empty index set $C=\{c_{j}|c_{j}\in \left\{0,1,\ldots ,m-1\right\}\}$ and a labeling scheme L of X, by replacing l(x) with $l^{C}\left(x\right)$ for all x∈X, the resultant labeling scheme is a PLS of X, denoted by $L^{C}$.*


Note that in PLS $L^{C}$, $2^{\left|C\right|}$ distinct labels are mapped to $2^{m}$ constellation points, where |C| obtains the cardinality of C. Therefore, each label is mapped to $2^{m-|C|}$ points.

**Definition** **3.**
***(PGLS)***
*: If a PLS $L^{C}$ satisfies the Gray labeling rule, it is called a PGLS. Here, the Gray labeling rule means that for any pair of constellation points $\left(x_{i},x_{j}\right),x_{i},x_{j}\in X,i\neq j$, which have the minimum Euclid distance $d_{min}$ in X, their PLs $l^{C}\left(x_{i}\right)$ and $l^{C}\left(x_{j}\right)$ have at most one bit in difference.*


Note that, since each label in PLS $L^{C}$ is mapped to $2^{m-|C|}$ points, there might be pairs of constellation points $\left(x_{i},x_{j}\right),x_{i},x_{j}\in X,i\neq j$, which have the minimum Euclid distance $d_{min}$ in X, which have the same PLs. In this case, PLS $L^{C}$ still satisfies the Gray labeling rule as explained in Definition 3. Therefore, $L^{C}$ is a PGLS.

In [1], it was conjectured that Gray labeling maximizes the BICM capacity. Based on this principle, we propose the criterion for optimal labeling schemes of a given delay pattern next.

#### 4.1.2. Criterion for Optimal Labeling Schemes

It can be seen from Equation (Equation 5) that the DBICM capacity is the sum of the bit channel capacities of delayed bits $l_{A}$, i.e., $\sum_{i\in A}I\left(l_{i};y\right)$, and the bit channel capacities of undelayed bits $l_{A^{c}}$, i.e., $\sum_{j\in A^{c}}I\left(l_{j};y|l_{A}\right)$. To optimize the DBICM capacity, the capacities of delayed and undelayed bits should be maximized simultaneously.

For the delayed bits $l_{A}$, their bit channel capacities are equal to those in BICM. Based on the Gray labeling rule for BICM, the PLS for C=A, i.e., $L^{A}$, should be a PGLS in order to maximize $\sum_{i\in A}I\left(l_{i};y\right)$.

For the undelayed bits $l_{A^{c}}$, if the values of the delayed bits $l_{A}$ are given, i.e., $l_{A}=\left\{b_{0},\ldots ,b_{d}\right\}$, where d=|A|, their bit channel capacities are equal to the BICM capacities in the sub-constellation $X^{l_{A}}$. Here, $X^{l_{A}}=\left\{x|x\in X,l^{A}\left(x\right)=l_{A}=\left\{b_{0},\ldots ,b_{d}\right\}\right\}$. Therefore, in order to maximize the channel capacity of undelayed bits, i.e., $\sum_{j\in A^{c}}I\left(l_{j};y|l_{A}\right)$, the labeling of $l_{A^{c}}$ should satisfy the Gray labeling rule in the sub-constellations $X^{l_{A}}$ for all $l_{A}\in {\left\{0,1\right\}}^{d}$.

In summary, the criterion for optimal labeling schemes is as follows:


***Criterion 1:***
The PLS of delayed bits $l_{A}$, i.e., $L^{A}$, should be a PGLS.The labeling of undelayed bits $l_{A^{c}}$ should satisfy the Gray labeling rule in the sub-constellations $X^{l_{A}}$ for all $l_{A}\in {\left\{0,1\right\}}^{d}$.


Let $L_{11}$ be the set of labeling schemes that satisfy the first item of Criterion 1 and $L_{12}$ be the set of labeling schemes that satisfy the second item of Criterion 1, respectively. The set of candidate optimal labeling schemes is $L_{1}=L_{11}\cap L_{12}$. To obtain $L_{1}$, we can first find $L_{11}$ and then eliminate the labeling schemes L that are not in $L_{12}$, i.e., $\left\{L|L\in L_{11},L\notin L_{12}\right\}$.

Though Criterion 1 is optimal in the sense that the optimal label scheme should be in $L_{1}$, the search space, i.e., $\left|L_{1}\right|$, is still too large. To see this, let us consider the delay patterns in which one bit is delayed, i.e., |A|=1. We first examine the space of $L_{11}$ and then $L_{12}$. For |A|=1, for the delayed bits $l_{A}$, all possible PLSs $L^{A}$ are PGLSs since the PLs for A have only one bit. Therefore, the PLs of all constellation points have at most one bit in difference, which means they satisfy the Gray labeling rule. To find a labeling scheme $L_{11}\in L_{11}$, we first select 16 points $\{x_{i_{1}},x_{i_{2}},\ldots ,x_{i_{16}}\}\subset X$ arbitrarily and denote the sub-constellation by $X_{0}$. There are $C_{32}^{16}$ choices for $X_{0}$, where $C_{n}^{m}=n!/m!\left(n-m\right)!$ is the number of combinations of selecting *m* elements from *n* elements. Then, for each $x\in X_{0}$, let PL $l^{A}\left(x\right)=0$. For the undelayed bits $l_{A^{c}}$, $2^{|A^{c}|}=16$ distinct PLs will be assigned to $X_{0}$. There are $P_{16}^{16}$ ways to label $X_{0}$. For the other 16 points $x\in X,x\notin X_{0}$, denote them by $X_{1}$, and let their PLs $l^{A^{c}}\left(x\right)=1$. There are also $P_{16}^{16}$ ways to label the undelayed bits of $X_{1}$. Therefore, $|L_{11}|=C_{32}^{16}\times P_{16}^{16}\times P_{16}^{16}$. Turning to $L_{12}$, we cannot calculate the exact number of labeling schemes that satisfy Criterion 1(b). However, our simulations showed that $L_{12}$ could not be found in a reasonable time frame. Therefore, by only considering Criterion 1 for optimal labeling schemes, the size of $L_{1}$ is still too large.

In order to reduce the search space, we propose more criteria for good labeling schemes in the following.

#### 4.1.3. Criteria for Good Labeling Schemes

***Criterion 2***: The PLS $L^{A^{c}}$ of the undelayed bits $l_{A^{c}}$ should be a PGLS.   

Note that Criterion 2 is different from the second item of Criterion 1. In Item 2 of Criterion 1, the labeling of undelayed bits is required to satisfy the Gray labeling rule only in the sub-constellations. However, in Criterion 2, the PLS $L^{A^{c}}$ of the undelayed bits should satisfy the Gray labeling rule in the whole constellation so that it is a PGLS.

Denote $L_{2}$ as the set of labeling schemes that satisfy Criterion 2. It can be seen that $L_{2}\subseteq L_{12}$ since if a labeling scheme satisfies Criterion 2, it must satisfy Item 2 of Criterion 1, but the inverse is not true. Therefore, the search space for good labeling schemes is reduced by replacing Item 2 of Criterion 1 with Criterion 2, i.e., $|L_{2}\cap L_{11}\left|\leq \right|L_{12}\cap L_{11}|$. Note that by introducing Criterion 2, the optimal labeling scheme may not be found since the optimal labeling scheme is guaranteed to be within $L_{12}\cap L_{11}$, but not guaranteed to be within $L_{2}\cap L_{11}$. Therefore, we say that Criterion 2 is for a good labeling scheme instead of for the optimal labeling scheme. However, the simulation results in Section 5 showed that the obtained good labeling scheme could almost approach the constellation capacity at a mid-to-high information rate regime.

Though the proposed Criterion 2 reduced the search space for good labeling schemes, it was still too large. To see this, let us consider the delay patterns in which two bits are delayed as an example. For |A|=2, in order to find $L_{2}$, we first search for all PLSs $L^{A^{c}}$ that satisfy Criterion 2. Note that for each PLS $L^{A^{c}}$, $2^{|A^{c}|}=8$ unique PLs are assigned to the 32 constellation points, and each of the unique PLs are assigned to $2^{\left|A\right|}=4$ constellation points. Then, for each $L^{A^{c}}$, the full labeling schemes are constructed by assigning the labels of the delayed bits to the four points with the same PL. For each group of four points with the same PL, there are $P_{4}^{4}=24$ ways to label the delayed bits. By considering that there are eight groups of points with different PLs for the delayed bits, each $L^{A^{c}}$ can generate ${\left(P_{4}^{4}\right)}^{8}$ full labeling schemes. We found the number of PLSs $L^{A^{c}}$ that satisfy Criterion 2 by simulations. It was approximately equal to $1.43\times 10^{8}$. Therefore, $|L_{2}|\approx 1.43\times 10^{8}\times {\left(P_{4}^{4}\right)}^{8}$ for |A|=2. To find $L_{11}$, we first found all PLSs $L^{A}$ that satisfied Item 1 of Criterion 1. Then, for each $L^{A}$, the full labeling schemes could be constructed in the same way as that for finding $L_{2}$. Finally, we found that $|L_{11}|\approx 3.7\times 10^{9}\times {\left(P_{8}^{8}\right)}^{4}$. Obviously, the sizes of $L_{2}$ and $L_{11}$ were still very large. In order to further reduce the search space, we propose the following criterion.

***Criterion 3***: For a good labeling scheme L, there exists an index set $A^{*}=\left\{i_{1},i_{2},i_{3},i_{4}\right\}\subset \left\{0,1,\ldots ,4\right\}$, and PLS $L^{A^{*}}$ is a PGLS.   

Denote $L_{3}$ as the set of the labeling that satisfies the Criterion 3. Now, our target for obtaining the set of good labeling candidates is to find $L=L_{11}\cap L_{2}\cap L_{3}$. To show that Criterion 3 must be satisfied by a good labeling, we will prove that the following proposition holds.

**Proposition** **1.**
*For each undelayed bit $l_{j},j\in A^{c}$, its channel capacity in DBICM is larger than or equal to its channel capacity in BICM, i.e.,*
(6)$$I\left(l_{j};y\right)\leq I\left(l_{j};y|l_{A}\right).$$


**Proof.** The left-hand-side of Equation (Equation 6) can be written as:
(7)$$\begin{array}{cc} \; & I(l_{j};y|l_{A})\\ \; & =H(l_{j}|l_{A})-H\left(l_{j}|y,l_{A}\right)\\ \; & =H\left(l_{j}\right)-\left(H\left(l_{j}\right)-H\left(l_{j}|l_{A}\right)\right)-H\left(l_{j}|y,l_{A}\right)\\ \; & =H\left(l_{j}\right)-I(l_{j};l_{A})-H\left(l_{j}|y,l_{A}\right). \end{array}$$
where H(·) is the comentropy function. In both DBICM and BICM, for using an ideal bit-interleaver, all bits are independent, i.e., $I(l_{i};l_{j})=0$ for all i,j∈{0,…,m−1} and i≠j. Therefore, $I(l_{j};l_{A})=0$ and Equation (Equation 7) can be written as:
(8)$$\begin{array}{cc} \; & I(l_{j};y|l_{A})\\ \; & =H\left(l_{j}\right)-H\left(l_{j}|y,l_{A}\right)\\ \; & =H\left(l_{j}\right)-H\left(l_{j}|y\right)+H\left(l_{j}|y\right)-H\left(l_{j}|y,l_{A}\right)\\ \; & =I\left(l_{j};y\right)+I\left(l_{j};l_{A}|y\right)\geq I\left(l_{j};y\right). \end{array}$$Note that $I(l_{j};l_{A})=0$ does not imply $I(l_{j};l_{A}|y)=0$ since *y* can be regarded as a function of $l_{j}$ and $l_{A}$. When *y* is known, the MI between $l_{j}$ and $l_{A}$ depends on the structure of *y*. □

Based on Proposition 1 and Equation (Equation 5), the following proposition holds.

**Proposition** **2.**
*For a given constellation X, labeling scheme L, and SNR, the DBICM capacity is larger than or equal to the BICM capacity, i.e.,*
(9)$$\begin{array}{cc} C_{DBICM} & =\sum_{i\in A}I\left(l_{i};y\right)+\sum_{j\in A^{c}}I\left(l_{j};y|l_{A}\right)\\ \; & \geq \sum_{i\in A}I\left(l_{i};y\right)+\sum_{j\in A^{c}}I\left(l_{j};y\right)\\ \; & =C_{BICM} \end{array}$$


It can be seen from Proposition 2 that the BICM capacity is the lower bound of the DBICM capacity. If a labeling scheme satisfies Criterion 3, it can maximize the BICM capacities of four bits and expect to have a high BICM capacity. Therefore, Criterion 3 can optimize the lower bound of the DBICM capacity and should be satisfied by a good labeling scheme.

### 4.2. Good Labeling Search Guideline

The proposed criteria for good labeling are summarized in Table 1. To find the candidate labeling set $L=L_{11}\cap L_{2}\cap L_{3}$, one can find the set with the minimum number of elements in $L_{11},L_{2}$, and $L_{3}$ first. Then, the labeling schemes in the smallest set are checked against the other two criteria. The labeling that does not belong to *L* is eliminated.

In this paper, we find $L_{3}$ first. To achieve this, all the four bit PGLS $L^{A^{*}}$ are enumerated and are denote by $L^{*}$. Then, the full labeling schemes are constructed from $L^{*}$, and $L_{3}$ is obtained. Then, the labeling that does not belong to $L_{11},L_{2}$ is eliminated from $L_{3}$.

Note that with $L_{3}$, various delayed patterns should be considered. For each delay pattern, all labeling schemes in $L_{3}$ are checked against Item 1 of Criteria 1 and Criterion 2. Take the delay patterns with one delayed bit as an example. For each labeling $L\in L_{3}$, there are $C_{5}^{1}$ delay patterns $D_{i}=\left\{T_{i}=1,\left\{T_{j}=0\right\},j=\left\{0,1,\ldots ,m-1\right\}\setminus i\right\},i\in \left\{0,1,\ldots ,m-1\right\}$ to be considered. If no delay patterns can let the labeling L satisfy Item 1 of Criteria 1 and Criterion 2 simultaneously, L is eliminated from $L_{3}$. With this process, all the candidate labeling schemes L∈L and the associated delay patterns are obtained. Then, for each L∈L and the associated delay patterns, the DBICM capacity is calculated. The labeling L and its associated delay pattern that achieve the highest DBICM capacity for a given SNR are considered to be the designed bit labeling scheme. In the following, an efficient three-step algorithm is proposed to implement this search guideline for good labeling schemes.

### 4.3. Search Algorithm for Good Labeling Schemes

In this section, we propose a three-step algorithm to find the candidate labeling schemes. In the PGLS search step, we find the set of 4 bit PGLSs $L^{*}$ by utilizing an efficient depth-first search algorithm. In the full labeling generation step, we utilize the PGLSs $L^{A^{*}}\in L^{*}$ to construct the full labeling. The final step is to eliminate the labeling that does not belong to $L_{1a},L_{2}$ simultaneously. Furthermore, we eliminate some strictly equivalent labeling schemes (SELS), which are defined as below, from the candidate set.

For two DBICM labeling schemes $L_{1}$ and $L_{2}$ of a constellation X, the sets of delayed and undelayed bits are $A,A^{c}$, respectively. The partial labels of a point *x* for A associated with $L_{1}$ and $L_{2}$ are denoted by $l_{\left(1\right)}^{A}\left(x\right),l_{\left(2\right)}^{A}\left(x\right)$, respectively. The partial labels of a point *x* for $A^{c}$ associated with $L_{1}$ and $L_{2}$ are denoted by $l_{\left(1\right)}^{A^{c}}\left(x\right),l_{\left(2\right)}^{A^{c}}\left(x\right)$, respectively. Let ${\left|y,z\right|}_{H}$ be the Hamming distance between two (partial) labels *y* and *z* of *x*.

**Definition** **4.**
***(SELS)***
*: If for any pair of constellation points $x_{1},x_{2}\in X$, $|\left(l_{\left(1\right)}^{A}\left(x_{1}\right),l_{\left(1\right)}^{A}\left(x_{2}\right)\right){|}_{H}={\left|\left(l_{\left(2\right)}^{A}\left(x_{1}\right),l_{\left(2\right)}^{A}\left(x_{2}\right)\right)\right|}_{H}$ and $|\left(l_{\left(1\right)}^{A^{c}}\left(x_{1}\right),l_{\left(1\right)}^{A^{c}}\left(x_{2}\right)\right){|}_{H}={\left|\left(l_{\left(2\right)}^{A^{c}}\left(x_{1}\right),l_{\left(2\right)}^{A^{c}}\left(x_{2}\right)\right)\right|}_{H}$ are satisfied simultaneously, $L_{1},L_{2}$ are called SELS (an example of the SELS is shown in Figure 3.*


Note that for a pair of SELS $L_{1}$ and $L_{2}$, their DBICM capacities are the same. This is because with $|\left(l_{\left(1\right)}^{A}\left(x_{1}\right),l_{\left(1\right)}^{A}\left(x_{2}\right)\right){|}_{H}={\left|\left(l_{\left(2\right)}^{A}\left(x_{1}\right),l_{\left(2\right)}^{A}\left(x_{2}\right)\right)\right|}_{H}$, the capacities of the delayed bits for $L_{1},L_{2}$ are the same, and with $|\left(l_{\left(1\right)}^{A^{c}}\left(x_{1}\right),l_{\left(1\right)}^{A^{c}}\left(x_{2}\right)\right){|}_{H}={\left|\left(l_{\left(2\right)}^{A^{c}}\left(x_{1}\right),l_{\left(2\right)}^{A^{c}}\left(x_{2}\right)\right)\right|}_{H}$, the capacities of the undelayed bits for $L_{1},L_{2}$ are the same. Based on (5), $L_{1}$ and $L_{2}$ have the same DBICM capacity. In addition, $L_{1}$ and $L_{2}$ have the same BER performance since for any pair of constellation points, the correspondence between the Hamming distance of their labels and their Euclidean distance remains the same.

The derivation of the three-step algorithm is shown in the Appendix A.

### 4.4. Capacity Equivalent Labeling Schemes

Although we excluded the SELS in the final step of constructing *L*, there are still some labeling schemes that have the same DBICM capacity, but are not SELS. We call these labeling schemes capacity equivalent labeling schemes (CELS). We investigate CELS since though they are equivalent in the sense of DBICM capacity, they have different BER performance. In this work, we aim to find the labeling scheme that not only has a high DBCIM capacity, but also a good BER performance. In the following, CELS is defined.

**Definition** **5.**
***(CELS)***
*: For two DBICM labeling schemes $L_{1}$ and $L_{2}$ with the same PLS of delayed bits, i.e., $L_{1}^{A}=L_{2}^{A}$. For each sub-constellation $X^{l_{A}}$ determined by the values of delayed bits $l_{A}$, if for any pair of constellation points $x_{1},x_{2}\in X^{l_{A}}$, $|\left(l_{\left(1\right)}^{A^{c}}\left(x_{1}\right),l_{\left(1\right)}^{A^{c}}\left(x_{2}\right)\right){|}_{H}={\left|\left(l_{\left(2\right)}^{A^{c}}\left(x_{1}\right),l_{\left(2\right)}^{A^{c}}\left(x_{2}\right)\right)\right|}_{H}$ holds. The two labeling $L_{1},L_{2}$ are called CELS.*


It can be seen from Equation (Equation 5) that the capacity of undelayed bits is the average of the BICM capacities in each sub-constellation $X^{l_{A}}$. A pair of CELS has the same BICM capacity in each sub-constellation $X^{l_{A}}$; therefore, they have the same capacity of undelayed bits. Their capacities of delayed bits are also the same since they have the same PLS of delayed bits. To sum up, a pair of CELS has the same DBICM capacity.

Note that a pair of SELS is a pair of CELS, but the inverse is not necessarily true. For example, two labeling schemes $L_{1},L_{2}$ are shown in Figure 4a,b, respectively. The delayed bits are $A=\{l_{0},l_{1}\}$. The undelayed bits are $A^{c}=\left\{l_{2},l_{3},l_{4}\right\}$, which are in the parentheses. The two labeling have the same PLS of delayed bits, i.e., $L_{1}^{A}=L_{2}^{A}$. Each eight points with the same color constitute a sub-constellation $X^{l_{A}}$. It can be seen that, for any two points $x_{1},x_{2}\in X^{l_{A}}$, $|\left(l_{\left(1\right)}^{A^{c}}\left(x_{1}\right),l_{\left(1\right)}^{A^{c}}\left(x_{2}\right)\right){|}_{H}={\left|\left(l_{\left(2\right)}^{A^{c}}\left(x_{1}\right),l_{\left(2\right)}^{A^{c}}\left(x_{2}\right)\right)\right|}_{H}$ holds. Therefore, the two labeling $L_{1},L_{2}$ are CELSs. In Figure 4a, the labels of the two points in the box are 10(000),11(100), and the Hamming distance of the undelayed PL is one. In Figure 4b, the labels of the same two points are 10(000),11(000), and the Hamming distance of the undelayed PL is 0. Therefore, $L_{1},L_{2}$ are not SELS, and they will have different BER performance.

The calculation of DBICM capacity in Equation (Equation 5) is based on the hypothesis that all the delayed bits are perfectly known by the demapper. This holds in a high SNR region. However, in a low SNR region, the hypothesis is not valid. Therefore, if the delayed bits have errors, the demodulation structure of DBICM for the undelayed bits is destroyed, i.e., we cannot obtain the full DBICM capacities of the undelayed bits in the wrong sub-constellation. The demodulation results depend on the received value and the labeling of the whole constellation.

In this paper, we only consider the minimum Euclidean distance $d_{min}$ in the constellation as the error radius, which is held in most instances. This is because the probability of making decision errors from one point to another point decreases exponentially w.r.t the Euclidean distance between two points. In the 32QAM constellation, for a signal *s* mapped by constellation point *x*, if the error happens in the transmission process, the received value *y* can be demodulated to the neighbor constellation points, which have the distance $d_{min}$ away from *x*. The cost of this error is the Hamming distance of the labels between *x* and its neighbor. Therefore, we employ the sum Hamming distance to evaluate the CELS, which is the total cost of the errors with a radius of $d_{min}$.

The labeling schemes $L_{1}$ and $L_{2}$ in Figure 4 are CELS, and the sum Hamming distances for $L_{1}$ and $L_{2}$ are 68 and 60, respectively. Therefore, $L_{2}$ is expected to have a better BER performance in a low SNR regime than $L_{1}$. In a high SNR regime, $L_{1},L_{2}$ will have almost the same BER. Numerical results in Figures 7 and 8 will confirm this expectation.

Note that we focus on the 32-QAM constellation in this paper for its bandwidth efficiency and wide use in practical systems [25,26,27,28], e.g., optical communication. The proposed method can also be extended to higher order modulation such as 128-QAM and 512-QAM, but the complexity becomes prohibitively high. The reason is that our search method is based on the algebraic structure of 32-QAM. For 128-QAM and 512-QAM, there is an exponential growth of the constellation points. Therefore, applying our method to these constellations will lead to the exponential growth of the complexity. For example, the numbers of candidate labeling schemes associated with a PGLS in 128-QAM and 512-QAM are $2^{63}$ and $2^{255}$, respectively. Contrary to the number of $2^{15}$ in 32-QAM, the search spaces of 128-QAM and 512-QAM are too large.

## 5. Numerical Results

In this section, we first show our designed labeling schemes with the associated delay patterns. Then, the DBICM capacities of these labeling schemes are compared to the constellation capacity and that of the quasi-Gray labeling in [10]. BICM capacity with the quasi-Gray labeling is also demonstrated for comparison. Finally, we show the BER performance of LDPC coded DBICM and BICM with the proposed labeling schemes and the quasi-Gray labeling scheme.

### 5.1. The Proposed Labeling Schemes

We utilize the proposed labeling search algorithm in Section 4.4 to find the candidate labeling schemes that have a high DBICM capacity. In the 4 bit PGLS search step, $|L^{*}|=$ 174,792 4 bit PGLSs are found. By executing the full labeling generation step and eliminating the SELS, $|L_{3}|$ = 33,292,288 candidate labeling schemes are found. For the delay patterns with 1,2,3, or 4 bits delayed by one time slot, the numbers of candidate labeling are |L|= 7,891,148, 16,683,242, 16,683,242, and 7,891,148, respectively.

Then, the DBICM capacities for these labeling schemes with their delay patterns at the SNR of 7 dB and 9.2 dB were calculated. We considered this SNR since the DBICM channel capacity was around 2.5, three bits/symbol or the code rate was around 0.5, 0.6 at this point, respectively, which is practical for high order coded modulation schemes. To eliminate the CELS introduced in Section 4.4, the sum Hamming distances of these labeling schemes were calculated. The one with the minimum sum Hamming distance was selected as our designed labeling scheme.

Our designed labeling scheme for the delay patterns with 1,2, and 3 bits delayed was the same, which is shown in Figure 4b. The designed labeling scheme for the delay patterns with 4 bits delayed is shown in Figure 5. The associated delay patterns were $D_{1}=\left\{T_{0}=T_{1}=T_{3}=T_{4}=0,T_{2}=1\right\}$, $D_{2}=\left\{T_{0}=T_{1}=1,T_{2}=T_{3}=T_{4}=0\right\}$, $D_{3}=\left\{T_{0}=T_{1}=0,T_{2}=T_{3}=T_{4}=1\right\}$, and $D_{4}=\left\{T_{1}=T_{2}=T_{3}=T_{4}=1,T_{0}=0\right\}$, respectively.

We also found the optimal delay patterns for the quasi-Gray labeling by enumerating all possible delay patterns. The associated delay patterns were $D_{1}^{\prime }=\left\{T_{1}=T_{2}=T_{3}=T_{4}=0,T_{0}=1\right\}$, $D_{2}^{\prime }=\left\{T_{0}=T_{2}=1,T_{1}=T_{3}=T_{4}=0\right\}$, $D_{3}^{\prime }=\left\{T_{3}=T_{4}=0,T_{0}=T_{1}=T_{2}=1\right\}$, and $D_{4}^{\prime }=\left\{T_{1}=T_{2}=T_{3}=T_{4}=1,T_{0}=0\right\}$, respectively.

### 5.2. The Channel Capacity

The DBICM channel capacities with our designed labeling schemes and the quasi-Gray labeling scheme are shown in Figure 6. The BICM channel capacity of the quasi-Gray labeling scheme is also shown in Figure 6 for comparison.

It can bee seen from Figure 6 that for the same number of delayed bits, our designed labeling scheme outperformed the quasi-Gray labeling scheme in terms of DBICM capacity. It is notable that for our designed labeling scheme with three bits delayed by one time slot, the DBICM capacity was only 0.015 dB away from the constellation capacity at an information rate greater than 2.5 bits/symbol.

### 5.3. BER Performance

In this subsection, we evaluate the BER performance of LDPC coded DBICM and BICM with various labeling schemes. For the DBICM capacity of 2.5bits/symbol, an LDPC code with code length n= 10,000 and code dimension k=5000 was constructed by the progressive edge-growth (PEG) algorithm [29], where the distribution of the variable node degree is optimized by the differential evolution algorithm [30]. For the DBICM capacity of three bits/symbol, the LDPC code from DVB-S2 [31] with the code rate 3/5 and code length 64,800 was employed. The interleaver used the consecutive mapping [3] in the coded modulation. Figure 7 and Figure 8 show the BER performance of the DBICM capacities of 2.5 bits/symbol and three bits/symbol, respectively. We can see that our designed labeling schemes had better BER performance compared to that of quasi-Gray labeling.

In Figure 7 and Figure 8, we also compare the labeling schemes $L_{1},L_{2}$ in Figure 4a,b to show the difference of the BER of CELS. We can see that in the low SNR region, $L_{1}$ had worse BER performance than that of $L_{2}$. In the high SNR regime, $L_{1}$ had nearly the same BER as $L_{2}$. This agreed with the expectation described in Section 4.4 that the one with the minimum Hamming distance would have better performance in the low SNR region.

## 6. Conclusions

In this paper, we considered the bit labeling design for 32-QAM DBICM. We aimed to obtain the labeling that had a high DBICM capacity and a good BER performance. To reduce the search space for the good labeling, three criteria were proposed. An efficient labeling search algorithm was proposed to obtain the labeling schemes that satisfied the proposed criteria. Then, we evaluated the candidate labeling with DBICM capacity in a fixed SNR point and selected the best one as our designed labeling. This SNR point must correspond to the most efficient capacity of the constellation. Capacity analysis showed that the designed labeling with three bits delayed by one time slot approached the 32-QAM constellation capacity within 0.015 dB at a mid-to-high information rate regime. BER simulation results also demonstrated that the designed labeling achieved a 0.1dB gain in terms of BER performance over the benchmark scheme.   

## Figures and Tables

**Figure 1 sensors-20-03528-f001:**
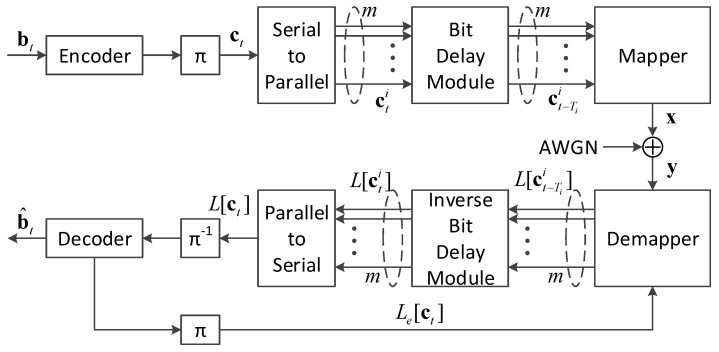
System model of a DBICM system.

**Figure 2 sensors-20-03528-f002:**
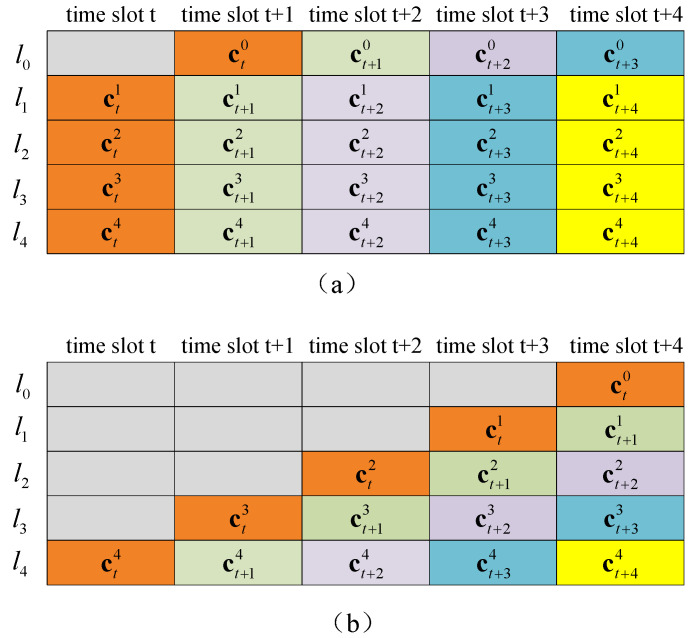
Two delay patterns for 32-QAM. (**a**) The delay pattern $D=\{T_{0}=1,T_{1}=T_{2}=T_{3}=T_{4}=0\}$. (**b**) The delay pattern $D=\{T_{0}=4,T_{1}=3,T_{2}=2,T_{3}=1,T_{4}=0\}$.

**Figure 3 sensors-20-03528-f003:**
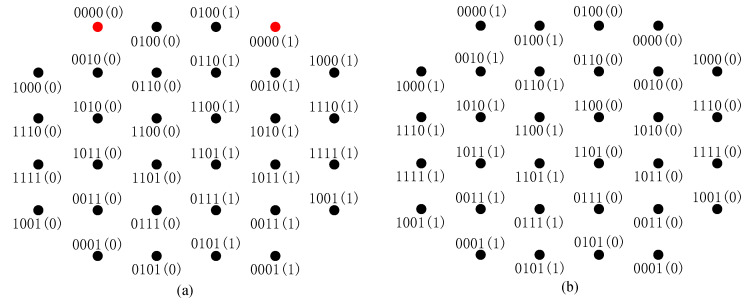
An example of the SELS. (**a**) Labeling scheme. $L_{1}$. (**b**) Labeling scheme $L_{2}$

**Figure 4 sensors-20-03528-f004:**
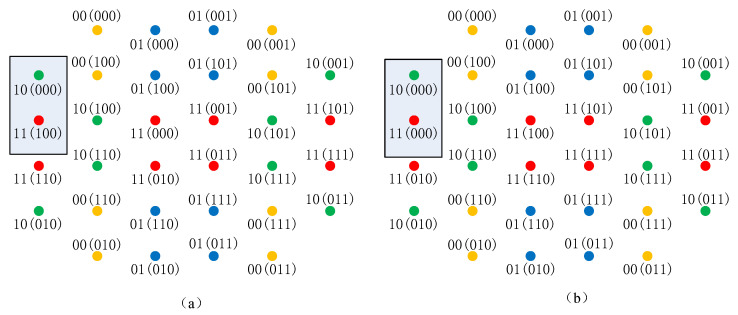
An example CELS. (**a**) The labeling $L_{1}$. (**b**) The labeling $L_{2}$.

**Figure 5 sensors-20-03528-f005:**
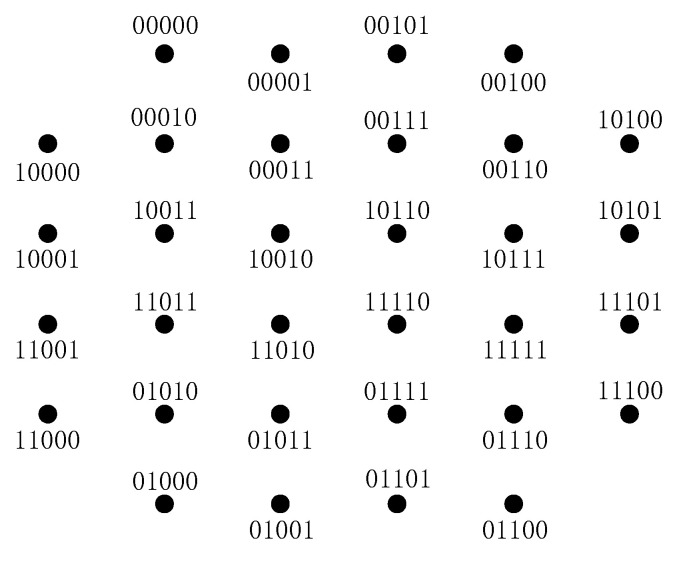
Designed labeling scheme with delayed pattern $D_{4}$.

**Figure 6 sensors-20-03528-f006:**
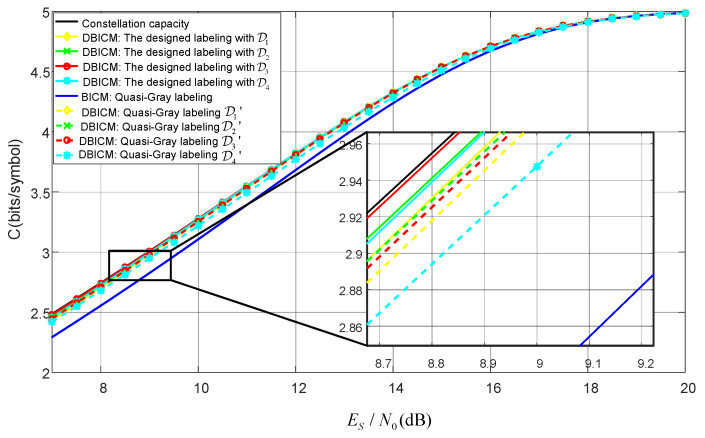
Channel capacity for our designed labeling and quasi-Gray labeling.

**Figure 7 sensors-20-03528-f007:**
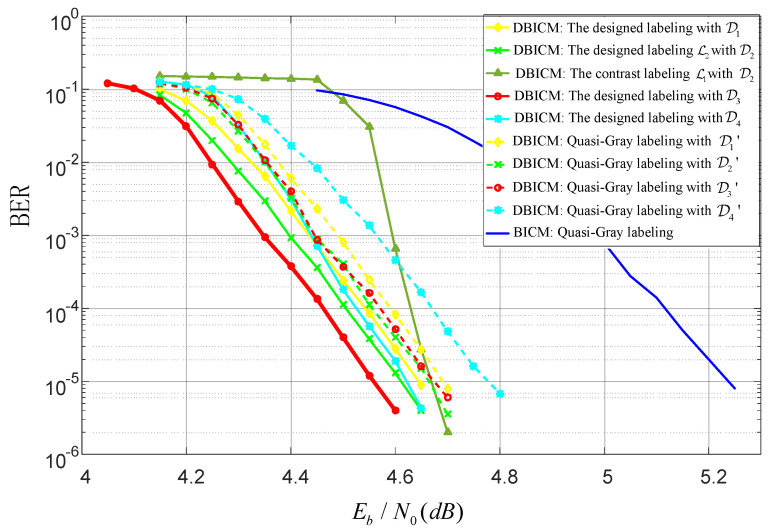
BER performance with code rate 0.5.

**Figure 8 sensors-20-03528-f008:**
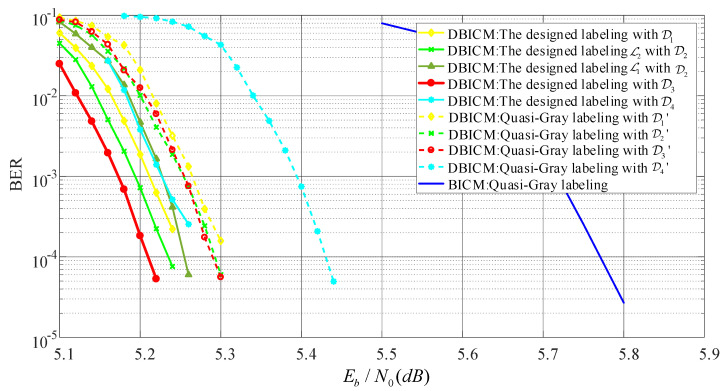
BER performance with code rate 0.6.

**Table 1 sensors-20-03528-t001:** The proposed criteria for good labeling.

Criteria Number	Description
1	The PLS of delayed bits $l_{A}$, i.e., $L^{A}$, should be a PGLS.
2	The PLS $L^{A^{c}}$ of the undelayed bits $l_{A^{c}}$ should be a PGLS.
3	PLS $L^{A^{*}}$ for which $A^{*}=\left\{i_{1},i_{2},i_{3},i_{4}\right\}\subset \left\{0,1,\ldots ,4\right\}$ is a PGLS.

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
