# Peer review of "Design of a Labeling Scheme for 32-QAM Delayed Bit-Interleaved Coded Modulation"

_sensors, 2020, doi:10.3390/s20123528_

Round 1

Reviewer 1 Report

Delayed bit-interleaved coded modulation (DBICM) is a coded modulation scheme proposed recently. This paper presents a labeling design for the DBICM with 32-QAM constellation. A three-step search algorithm is proposed to reduce the cost of the search process for the optimal labeling scheme. Numerical results are provided to illustrate the effectiveness of the proposed method. The paper is well written. I recommend it with a minor revision.

My other comments are as follows.

  1. The formats of references are not accordance, such as the 12th and 17th reference.
  2. The formats of abbreviations are not accordance, such as PGLS and Pls.
  3. There are some mistakes in the paper, such as the line 277, “A* \in L*” should be “L^{A*} \in L*”.
  4. The uses of \cdots and \ldots are not accordance throughout the paper. Please check carefully.
  5. Some uses of notation would cause confusion. For example, the set and vector are represented by the same boldface symbol. In the line 154, the index set \mathbf{c} is recommend to be \mathcal{C}.

Reviewer 2 Report

This paper proposes a labeling scheme for 32-QAM DBICM.

The following comments should be addressed.

1. The authors claimed that the labeling with a smaller sum Hamming distance is expected to has a better BER performance. Pleas complain this phenomen and give more results or proof for the proposed  sum Hamming distance. In addition,  the sum Hamming distance can be used in designing for the other QAMs for the DBCIM?

2. Could the proposed schemes  be extended into the 128-QAM or 512-QAM? This is to say, the proposed scheme is a general scheme or a special case? Please give more the extended methods and give more results.

3. The authors use the old reference [7] published in 1997 as the benchmark. In the 20 years, this topic is not  further studied? This reviewer thinks that the authors should give more discussions on the BICM or DBCIM in the introduction. It is very important that the authors should give more comparisons with the existing schemes.

Reviewer 3 Report

This paper proposes a new labeling scheme for 32-QAM delayed bit-interleaved coded modulation (DBICM). The main contribution is the proposal of an algorithm to find all candidate labeling schemes based on a proposed criterion that uses the Gray labeling rule

      The following contents need to be improved:

In line 15 it is presented reference 1 and then it jumps to 13-16, and the reference 2 is presented until line 27.  I recommend to present all the references in numerical order.

In line 250 an extra character appears at the end of the sentence.

I suggest adding a table that contains a summary of the criteria for good labeling schemes. In this way, readers can better guide themselves to differentiate procedures.

In line 385 replace “form” for “from”.

In the numerical results section, the authors compare their proposal to the reference [7], it could be interesting to add a comparison with newer works, such as:

D. Wesel, Xueting Liu, J. M. Cioffi and C. Komninakis, "Constellation labeling for linear encoders," in IEEE Transactions on Information Theory, vol. 47, no. 6, pp. 2417-2431, Sept. 2001, doi: 10.1109/18.945255.

D. Wesel and Xueting Liu, "Edge profile optimal constellation labeling," 2000 IEEE International Conference on Communications. ICC 2000. Global Convergence Through Communications. Conference Record, New Orleans, LA, USA, 2000, pp. 1198-1202 vol.3, doi: 10.1109/ICC.2000.853689.

Use capital letters on the line 458 “Conclusion”.

In the conclusions, the authors should explain if there are restrictions for the proposed method, furthermore, the main achievements obtained by implementing the proposed method should be emphasized.

Reviewer 4 Report

The paper describes an algorithm to obtain a labeling scheme for the relatively novel concept of delayed BICM.

Positives:

The paper clearly explains the concept of DBICM and its potential benefits w.r.t. standard BICM. The three proposed design criteria are sounds and well justified. The proposed improvements of 0.1 dB are significant within the context of 32QAM. 

Negatives that could be addressed in a revision:

1/ The algorithm is a bit lengthy in derivation and hard to follow. Consider moving to appendix. Due to its complexity, I believe that sharing of the code would be much appreciated for the audience.

2/ The main problem that I see with the paper is its limitation to 32QAM. It can be broken down into the following issues:

2.1/ It is not discussed if the algorithm generalizes to larger modulation formats. Is it still applicable in terms of performance and running time?

2.2/ For standard larger and smaller modulation formats (e.g. 16QAM and 64QAM), well established multi-level coding (MLC) schemes exist which achieve capacity. MLC with 2-stage decoding will have the same complexity as DBICM with maximum delay of 1 period, and in the 16QAM case will actually be capacity achieving. I suggest to compare both the capacity and BER results with such an MLC scheme. At least the capacity curves should be easy to obtain for standard 16QAM and 64QAM MLC with set-partitioning labeling. 

Observe that at 2.5 bits/symbol, 16QAM capacity is very close to 32QAM capacity, so the optimal MLC with 2-stage decoding is likely more relevant than the proposed 32QAM DBICM with the same complexity.

3/ I suggest the paper to go through a language proof service. While the points made are clear and understandable, language mistakes often make the text annoying to read. 
